# Robotic-Assisted Surgery for Primary Hepatobiliary Tumors—Possibilities and Limitations

**DOI:** 10.3390/cancers14020265

**Published:** 2022-01-06

**Authors:** Julia Spiegelberg, Tanja Iken, Markus K. Diener, Stefan Fichtner-Feigl

**Affiliations:** Department of General and Visceral Surgery, University Medical Center Freiburg, Hugstetterstraße 55, 79106 Freiburg, Germany; julia.spiegelberg@uniklinik-freiburg.de (J.S.); tanja.iken@uniklinik-freiburg.de (T.I.); stefan.fichtner@uniklinik-freiburg.de (S.F.-F.)

**Keywords:** robotic surgical procedures, liver neoplasms, hepatocellular carcinoma, biliary tract neoplasms, minimally invasive surgical procedures, robotic liver hepatectomy, robotic liver resection

## Abstract

**Simple Summary:**

Primary liver malignancies are some of the most common and fatal tumors today. Robotic-assisted liver surgery is becoming increasingly interesting for both patients and surgeons alike. Up to date, prospective comparative studies around the topic are scarce. This leads us to an ever existing controversy about the efficacy, safety, and economic benefits of robotic surgery as an extension of traditional minimally invasive surgery over open liver surgery. However, there is evidence that robotic-assisted surgery is, after passing the learning curve, equivalent in terms of feasibility and safety, and in some cases superior to traditional laparoscopic hepatic resection. With this work, we want to provide an overview of the latest and most significant reviews and meta-analyses focusing on robotic hepatectomy in primary liver malignancies. We outline the technical aspects of robotic-assisted surgery and place them into the context of technical, surgical, and oncological outcomes compared with laparoscopic and open resection. When chosen per case individually, any hepatic resection can be performed robotically to overcome limitations of laparoscopic surgery by an experienced team. In this paper, we propose that prospective studies are needed to prove efficacy for robotic-assisted resection in liver malignancy.

**Abstract:**

Hepatocellular and cholangiocellular carcinoma are fatal primary hepatic tumors demanding extensive liver resection. Liver surgery is technically challenging due to the complex liver anatomy, with an intensive and variant vascular and biliary system. Therefore, major hepatectomies in particular are often performed by open resection and minor hepatectomies are often performed minimally invasively. More centers have adopted robotic-assisted surgery, intending to improve the laparoscopic surgical limits, as it offers some technical benefits such as seven degrees of freedom and 3D visualization. The da Vinci^®^ Surgical System has dominated the surgical robot market since 2000 and has shown surgical feasibility, but there is still much controversy about its economic benefits and real benefits for the patient over the gold standard. The currently available retrospective case studies are difficult to compare, and larger, prospective studies and randomized trials are still urgently missing. Therefore, here we summarize the technical, surgical, and economic outcomes of robotic versus open and laparoscopic hepatectomies for primary liver tumors found in the latest literature reviews and meta-analyses. We conclude that complex robotic liver resections (RLR) are safe and feasible after the steep learning curve of the surgical team has plateaued. The financial burden is lower in high volume centers and is expected to decrease soon as new surgical systems will enter the market.

## 1. Introduction

Primary hepatobiliary cancer is the seventh most frequent cancer globally, with increasing incidence in recent decades in some areas like India, USA, and Europe and decreasing in other areas such as Asia. Hepatocellular carcinoma (HCC) is the most prevalent (90%) type of liver cancer, with the 3rd highest mortality rate worldwide. The most common causes are chronic Hepatitis B and C virus infections, alcohol and metabolic diseases [1].

Surgical treatment is the main component of primary hepatobiliary cancer management. Laparoscopic liver surgery, applied since 1993, has been technically challenging because of the anatomical traits of the liver, with its intensive and variant vascular and biliary system. More technically complex liver resections are accompanied by difficult accessibility of the vena cava, major hepatic veins, and the hilum, thus making conventional laparoscopy challenging and risky due to limitations of instrument movement, lack of depth perception, fixed fulcrum at the ports, and difficult suturing, particularly in the presence of hemorrhage. Therefore, complicated liver surgeries are still frequently performed with the open approach. In 2000, the United States Food and Drug Administration approved the da Vinci^®^ System (Intuitive Surgical, Inc., Sunnyvale, CA, USA), which is aimed to perform the techniques of open surgery in a minimally invasive approach by overcoming the obstacles of conventional laparoscopy. Over time, the company developed better versions of the da Vinci console to overcome some of its disadvantages. The most recent model is the da Vinci Xi marketed in 2017. The robot is operated by the surgeon sitting at a console. The EndoWrist technology instruments act as the surgeon’s hands allowing seven degrees of freedom dexterity (compared to 4 degrees in conventional laparoscopy) of the fine instruments enabling easier intracorporeal suturing and knotting, filtering fulcrum and tremors, while also providing high definition and micromotions [2]. Moreover, the robot provides 3-dimensional imaging, allowing better resolution, depth perception, and magnification and a stable camera platform for improved hand-eye coordination. Indocyanine green (ICG) fluorescence imaging is supported by the robotic platform. ICG can show lymphatic, biliary tract, and vascular structures, distinguish healthy and tumor tissue intraoperatively, and may help with the identification of resection lines, which could potentially reduce intraoperative complications [3,4,5]. Lastly, the ergonomic feature of sitting comfortably reduces the fatigue of the surgeon, which is important in difficult, long procedures [6].

Nevertheless, the robotic system faces challenges for it to be accepted on a large scale, especially its high costs and system-born disadvantages like the lack of tactile feedback. The cost–benefit effects are still controversial in the literature, resulting mainly from the fact that the volume of robotic liver surgery for HCC needed for randomized controlled trials is scarce [7].

In this narrative review, we firstly aim to provide an overview of robotic-assisted liver surgery in primary malignant liver tumors. We describe in this context interesting surgical, oncological and financial outcomes. Secondly, we present the current results on the learning curve in robotic liver surgery.

## 2. Methods

PubMed was searched using search terms like “robotic liver surgery”, “robotic liver resection” and “robotic liver hepatectomy”, in combination with “primary liver tumor”, “primary liver cancer”, “primary malignant liver tumors”, “hepatocellular carcinoma”, “cholangiocarcinoma”, as well as “costs”, “economic”, and “learning curve”. The primary search was further limited by selecting “reviews” and including the most recent publications since 2016 as well as older, but high-impact publications. Surgical techniques included minor and major resections (defined as resection of four more Couinaud segments) for primary liver neoplasms, defined as HCC or the less prevalent cholangiocellular carcinoma (CCC).

We have added a paragraph with descriptions of the technical approach to robotic liver resection, specifically right hemihepatectomy, according to our own center experience.

We established this narrative review as a summary of the current scientific consensus of data on robotic resections of primary liver tumors.

We have added Table 1 and Table 2 as a summary of the main primary analyses of the surgical outcome of minimally invasive resections of CCC and HCC patients, respectively. We do not aim for completeness in the sense of a systemic review.

### 2.1. Minimally-Invasive Techniques for Treatment of Primary Liver Tumors: Current Evidence

The most common primary malignant liver tumors are HCC followed by cholangiocellular carcinoma [16].

CCC is often diagnosed only at a locally advanced stage, which in many cases is associated with large tumors, a multifocal distribution pattern, and lymph node metastases, so that extensive liver resections are often necessary for the treatment of CCC and systematic lymphadenectomy is recommended [17].

Lymphadenectomy should also be performed for HCC in noncirrhotic liver due to the higher likelihood of lymph node metastases analogous to CCC [18]. In contrast, approximately 80% of HCCs occur in fibrotic and cirrhotic liver tissue. In these cases, lymphadenectomy is not recommended because of the significantly higher morbidity due to portal hypertension and the low risk for lymph node metastases.

Liver cirrhosis is also an important risk factor for MIS liver resection, as it is associated with a higher rate of decompensation of liver function (especially ascites production), longer stays in intensive and normal care units, and a higher 90-day overall mortality [19].

### 2.2. Cholangiocellular Carcinoma

There are only a few publications on minimally invasive techniques for treatment of CCC since it is a rare disease and the surgical therapy is in many cases more complicated than in other entities due to the biological particularities. Accordingly, the current evidence is based exclusively on retrospective (matched-pair) analyses with limited patient numbers [20].

In one of the largest series, 20 consecutive patients with MIS-operated CCC were matched 1:3 with corresponding open-operated patients. In this series, laparoscopic lymphadenectomy was performed in only 50% of patients compared with 97% in the open group, because standardized lymphadenectomy was not established until the second half of the study period and had also been routinely performed since then. The operative time after laparoscopic resection was significantly longer, but with less blood loss [8].

Wei et al. demonstrated in their series of 30 patients that a minimally invasive approach is technically feasible for both small and large and multifocal CCC. In this series, only selective lymphadenectomy was performed [21].

When comparing intrahepatic CCC (iCCC) with extrahepatic hilar CCC, the first observation is that MIS is considerably more common in iCCC [22].

Laparoscopic liver resections for iCCC are technically feasible and safe [23,24]. Regmi et al. demonstrated that LLR compared with OLR for iCCC is associated with lower blood transfusion rate, better R0/R1 resection rate and a shortened hospital stay. However, the authors note that in the laparoscopic technique group, there was a smaller tumor size and fewer major resections as well as a lower lymph node dissection rate performed. From an oncologic perspective, there are comparable 3-year overall survival and 5-year disease free survival [23].

Another meta-analysis agreed that patients with LLR for iCCC benefit in terms of short-term outcome, with long-term outcome being comparable [24].

When lymphadenectomy was performed laparoscopically, a median of eight lymph nodes were resected, complying with current AJCC treatment guidelines. Patients undergoing laparoscopic resection showed lower overall morbidity contributing at least in part to a decreased hospital stay [25].

The role of MIS for bismuth type I-III hilar CCC has increased significantly in recent years. In 2016, Xu et al. reported robotic resections including hemihepatectomy and lobus caudatus resection as well as trisectionectomy, lymphadenectomy, resection of extrahepatic bile duct, Roux-en-Y hepaticojejunostomy for Bismuth IV tumors. The authors note that significantly more complications occurred here due to the complexity of the resection, and the indication must be clearly stated [9].

A systemic review by Wang et al. summarized a total of 205 cases of minimally invasive resection of hilar CCC (35 cases Bismuth I, 22 cases Bismuth II, 68 cases Bismuth III, 13 cases Bismuth IV) [26]. The authors found technical feasibility and safety for minimally invasive resection of hilar CCC with an overall low conversion rate to open surgery (3.8% for RLR and 12.2% for LLR).

It has to be noted that robotics for hilar cholangiocarcinoma has been described simultaneously to pure laparoscopy, whereas its application to liver resections for intrahepatic malignancies has been documented much later than pure laparoscopy [27]. Overall, based on the scarce literature, MIS for hilar CCC is still in its infancy. Robotic resection and reconstruction seem to be a promising approach in this regard.

The advantages of robotics in surgery for CCC cannot primarily be measured by the usual parameters such as blood loss, operating time, surgical outcome, etc., but consist rather in technical aspects. It seems evident that the three-dimensionality and the flexibility of the surgical instruments considerably facilitate the dissection of the hepatic pedicle as well as the hepatic hilus during lymphadenectomy [8].

Sucandy et al. published a larger case series of robotic resection for extrahepatic CCC in 2021. The total of 15 patients showed no severe complications (grade 3 or higher according to Clavien–Dindo). They thus showed that robotic resection of extrahepatic CCC is safe, feasible, and associated with good clinical outcome. It should be noted here that the average tumor size was only 2 cm, an average of 3 lymph nodes were removed, and there was a total of 4 R1 resections (27%) [10].

## 3. Hepatocellular Carcinoma

Although most analyses do not differentiate between hepatocellular carcinoma (HCC) and cholangiocellular carcinoma (CCC), a very recent series from Sweden demonstrates that more than 80% of primary liver cancers are HCC and less than 20% are CCC [28]. Because no randomized trials of laparoscopic liver surgery for HCC are available at this point, matched-pair and meta-analyses represent the highest possible level of evidence. The most recent meta-analysis on the outcomes of minimally invasive surgery for HCC is based on 28 studies (1984 vs. 5245 patients). For the comparison of MIS with open resection for exclusively solitary HCC, 562 patients operated openly were compared with 441 patients operated minimally invasively. In this analysis, complication rate, blood loss, transfusion rate, and hospital stay were significantly lower after MIS resections. The minor resection subgroup (628 vs. 658 patients) also showed significant advantages for MIS in these parameters [29].

While operating times did not differ between laparoscopic and open resections for solitary HCC and minor resections, the operating time of open minor or anatomic HCC resections was significantly shorter (82 vs. 82 patients). However, major resections also showed a lower complication rate after MIS resection, whereas postoperative liver failure, perioperative mortality, and long-term survival did not differ [29]. In the subpopulation of older patients (>75 years), the incidence of ascites in particular, but also pulmonary and cardiovascular complications was significantly lower after MIS resection of HCC [30].

Laparoscopic liver resection, whenever feasible, seems to decrease blood loss, postoperative morbidity, and hospital stay compared to an open approach without hampering long-term oncological outcomes. These advantages have been acknowledged in the recent update of the EASL guidelines (European Association for the Study of the Liver) for the management of HCC [31].

Khan et al. performed a multicenter study with a total of 61 patients with HCC, CCC or gallbladder carcinoma who underwent robotic resection. A comparable long-term oncologic outcome was achieved compared with open or laparoscopic resection [32].

## 4. Robotic Resection of Primary Hepatic Tumors

Obviously, already known advantages of the minimally invasive technique over conventional open surgery seem to be confirmed for the robotically assisted technique as well [33].

However, the evidence for possible advantages and disadvantages compared to laparoscopic liver surgery is still low. The available publications are mostly case reports, small feasibility series, and isolated retrospective mono-center analyses, and sometimes pooled multicenter studies due to small patient numbers [14].

For clinical studies, experience has been gained, for example, in resection of perihilar cholangiocarcinoma [9] as well as in situ splits in the context of multimodal concepts (“associating liver partition and portal vein ligation”, ALPPS) [34].

The scientific value of existing publications usually does not go beyond a confirmation of “technical feasibility”, which by its nature is the first step in the implementation of a new technique. However, existing field reports seem to indicate that potential strategic advantages of the surgical robot are particularly apparent in highly complex resections, which would amount to an expansion of the minimally invasive surgical resections [35,36].

Intraoperative conversion to an open procedure is less common in robotic-assisted liver resections than in laparoscopic liver resections [37].

A large European bi-institutional study compared outcomes of liver resections for hepatobiliary malignancies between 111 laparoscopic and 61 robotic-assisted operations in a propensity score-matching study [38].

The authors found no differences in operative time, blood transfusion, and 90-day overall morbidity. A special focus was the oncologic outcomes, which indicated that the R0/R1 resection rate was similar in both groups. In addition, the 1-, 2-, and 3-year overall survival rates, as well as the 1-, 2-, and 3-year recurrence-free survival rates were similar between the laparoscopic and robotic cohorts. This study demonstrated that the excellent oncologic outcomes achieved with laparoscopic surgery can be maintained with robotic surgery.

For a more precise evaluation of possible advantages of robotic liver surgery, the comparison group must first be defined precisely. Compared to conventional open liver surgery, robotic surgery is convincing due to its lower invasiveness and thus significantly influences the short-term outcome [39].

However, if we compare the robotically assisted technique with the established minimally invasive techniques in liver surgery, possible differences are most likely to result from technical developments that are not currently available in laparoscopic surgery, at least not in sum. To what extent this will be reflected in an even further improved outcome of patients is currently unclear.

Randomized trials comparing laparoscopic and robotic liver surgery do not yet exist, likely due to the fact that both are approaches of minimally invasive surgery, but with different technical prerequisites. The most recent meta-analysis of available publications (n = 6; 345 RH vs. 748 LH) on this comparison found that robotic resections require a lower conversion rate but have a longer operation time and are more common for right hepatectomy.

On the other hand, there was no difference in estimated blood loss, blood transfusion rate, hospital stay, R0 resection rate, complications, resection margin, left lateral sectionectomy, and left hepatectomy [40].

However, it should be mentioned that in our opinion and experience the advantages of robotic surgery cannot be sufficiently evaluated by taking into account the usual comparison parameters (blood loss, perioperative complication rate, 90 days overall morbidity, length of hospital stay, etc.). The better accessibility of complex anatomical areas (e.g., segment VII/VIII) compared to the laparoscopic technique should certainly be mentioned. An advantage in lymphadenectomy with regard to a better overview would also be conceivable.

## 5. Technical Considerations

The surgical advantages of the surgical robot on the liver hilus are obvious. The 10× magnification and three-dimensionality allow precise resolution even in the pre-operated situs, which is unfamiliar from either laparoscopic or open surgery. In addition, there is the option of fluorescence imaging (Firefly Fluorescence Imaging, Intuitive Surgical, Sunnyvale, CA, USA), which, in addition to intraparenchymal orientation, also allows improved orientation in complex bile duct anatomy and thus offers advantages over laparoscopy.

In addition to state-of-the-art imaging, miniaturization and degrees of instrument motion with the filtering out of natural tremors should also be highlighted in this context. In combination, these technical developments allow uncomplicated suturing and reconstruction of structures. The lack of tactile “feedback” seems to be replaced by “visual haptics” and thus does not represent a major disadvantage [41]. The hepaticojejunostomy routinely performed as part of robotically assisted pancreatic head resections shows that the aforementioned advantages of the robotically assisted technique also make liver resection with biliary reconstruction conceivable [42].

Due to the technical advancements compared to laparoscopic techniques, a further improvement in the perioperative course as described above is not expected. Accordingly, possible advantages must be evaluated in a more differentiated approach. The evaluation of robotically assisted techniques concerning a possible increase in intraoperative safety and the expansion of the surgical spectrum to include highly complex liver resections, however, seems to make more sense [43]. The aforementioned advantages of the robotically assisted technique certainly result in differences in complex liver resection (Iwate score > 6), which may already represent a borderline area for laparoscopic liver surgery [44].

In this respect, the clinical significance of possible advantages of robotic-assisted liver surgery should not be judged solely based on the perioperative course, which is readily used for comparative purposes, but should be evaluated by taking into account the complexity of the procedure.

A steeper learning curve in comparison to laparoscopic liver surgery can be seen as a further advantage of the robotically assisted technique [45].

In addition to parenchymal dissection, crossing structures can also be preparated robotically assisted and the necessary vessel clips and staple sutures can be placed robotically assisted. However, the changeover times of robotic instruments and applicators are still significantly longer compared with the laparoscopic technique, which can be a clinically significant limitation, e.g., in the case of severe bleeding. Immediate treatment of bleeding is of superior importance due to the lack of good compression options in the minimally invasive technique. Consequently, technical solutions are needed that allow uncomplicated and rapid instrument changes, also on the surgical robot, to enable the console surgeon to operate independently.

Thus, currently, laparoscopic and robotic resection can be considered largely equivalent in quality.

## 6. Indication/Contraindication

While minimally invasive liver resection has shown many advantages including less blood loss, shorter hospital stay and better postoperative outcomes, the general indications for laparoscopic liver resections in the past were for small and peripheral tumors, and it was rarely adopted due to technical limitations. Those constraints were due to the intraoperative complex variability of the liver with its segments and its vascularity. Robotic minimally invasive surgery was developed to surpass these restrictions. In 2003, Giulianotti et al. [46] performed a segmental hepatic resection as the first robot-assisted laparoscopic liver procedure. The first experiences of robotic hepatectomy were wedge hepatectomy, hemihepatectomy, and extended hemihepatectomy. Simultaneously with a better understanding of the liver segments and improved imaging techniques, robotic systems offer an increased range of motion, improved visualization, and better ergonomics for the surgeon. Today, the indication, as well as the contraindication for robotic liver surgery, is the same as for laparoscopic liver surgery. In essence, any hepatobiliary procedure can be performed with the robot, however, most studies include mainly left lateral sectionectomy, left hemihepatectomy, right hemihepatectomy, anterior segmentectomies, and wedges resections, and the opinions of authors differ.

The main indication for RLR in liver malignancy is HCC, which is typical for portal vein tumor invasion and develops in a cirrhotic liver in over 80% of cases. Major hepatectomies made up 40% in a review of 255 RLRs. Thereafter, wedge resection or segmentectomies account for 37.7% and left lateral sectionectomy for 20.8% [47].

In 2008, an international consensus conference concluded the indications for laparoscopic liver surgery should be limited to solitary lesions smaller than 5 cm located in peripheral segments 2 to 6, with left lateral sectionectomy as the recommended procedure, while major laparoscopic liver resections (LLR) should only be performed by experienced surgeons. Preparation for conversion to an open procedure is beneficial. Good indications are also patients with first-time liver resection, with non-cirrhotic liver or Child’s A or B stage cirrhosis, esophageal varices grade ≥ 1 and a platelet count > 80 × 10^9^/L, American Society of Anesthesiologists grade ≥ 2 or grade ≥ 3 [48]. More and more studies are proposing indications for difficult segments of the liver for RLR. The first international consensus statement on robotic hepatectomy surgery was given in 2018, and stated that the number of robotic liver maneuvers and their indications is constantly growing [48,49].

Regarding indications for robotic liver surgery, it thus seems to be clear that oncological outcomes are comparable to those of open and laparoscopic approaches for HCC and CCC [12,50,51].

Contraindications, apart from those given for open hepatobiliary resection according to the international consensus statement in 2018, include no tolerance to pneumoperitoneum because of cardiopulmonary disease, intra-abdominal adhesions, tumors close to large blood vessels, invasion of hilum or major vessels, and tumors in need of extensive hilar lymph node dissection or biliary and vascular reconstruction. Age should not be a limitation to surgery for HCC [3,48,49]. Nowadays, for most authors, tumor size is not a contraindication for a robotic approach, rather the location of the tumor. Large tumors in the left lateral or pedunculated regions can be resected.

Tumors at the base of segment IV or V connected to the hilum or close to major hepatic veins, transection lines or extensive portal lymphadenectomy, as well as tumors located in posterior segments (I, VII and VIII), are contraindications, given the rupture risk, but these are not absolute anymore when robotic maneuvers are performed by highly experienced surgeons. However, if portal vein or vena cava replacement is necessary, an open approach is often still chosen [52].

It is important to note that no definitive indications and contraindications of robotic resection have been reported due to the lack of randomized control trials.

Currently, resection of major tumors with complex vascular reconstruction, as well as resection of perihilar cholangicarcinoma, are considered as relative contraindications and should be reserved for highly selected and experienced centers.

From a current technical point of view, biliary anastomoses and vascular resections with a robotic approach are feasible, as could be shown in pancreatic surgery [9]. Complex procedures like association liver partition and portal vein ligation for staged hepatectomy (ALPPS) can safely be performed with a robotic approach [34,53].

Nevertheless, there is a serious risk of tumor seeding and a high rate of locoregional recurrence after robotic perihilar cholangiocarcinoma resection [54].

## 7. Learning Curve

Complex liver surgery can be implemented robotically in centers with high expertise, with good short- and long-term outcomes. A meaningful learning curve to achieve the benefits of the robotic approach is necessary before performing major resections. The learning curve, in this case, is defined as the time to become proficient using the robotic system and completion of the operation without conversion to open surgery. The learning curve in robotic surgery involves the entire team, as skills like team communication have to be learned to achieve satisfying results [47]. Further, the learning curve in liver surgery is impacted by the complexity of the surgery, which differs significantly between cases, as well as the experience of the surgeon, making it difficult to compare. Furthermore, most studies are compared in a retrospective nature.

Chen et al. evaluated the learning curve of 183 robotic resections, including minor and major hepatectomies, for operation time and blood loss [55]. They proposed that the learning curve for robotic major resections is divided into three phases. The initial phase is completed after 15 surgeries and leads to a reduction in surgery duration and shortening of length of stay. Finally, after another 25 cases, blood loss can be reduced. In the final phase (52 cases), there is an overall improvement in robotic skills, which is comparable to the learning curve in laparoscopic hemihepatectomy. Thus, a stepwise approach is needed when introducing a robotic progression to avoid complications. With increased expertise, large exophytic growing tumors can eventually be operated on safely. Trocar positions and robot cart dock are subject to rapid improvement at the beginning of the learning curve. The different versions of the Da Vinci robotic surgical system are rarely reported in studies, but may also influence the learning curve, as the older systems are more difficult to handle, i.e., regarding docking and placing of ports. Ashraf et al. specifically researched the docking time and found the learning curve plateaued at 30 cases, thus arguing against the significance of docking time in operative time after the learning curve, especially when a newer Da Vinci model is used [56,57].

Moreover, Chen et al. reported their experience with 35 cases of robotic liver resection, 21 of which were left hemihepatectomies, and all of which were performed without table assistants. Previously, 300 surgeries had been performed with table assistance. As the volume of robotic resection is increasing and the learning curve with the robotic system is overcome, there is improvement documented in terms of intraoperative risks like blood loss [58]. Thus, they were able to demonstrate that the fully robotic approach is safe and feasible and leads to a significant reduction in surgical time. However, this interesting approach to make the console surgeon independent also has downsides in addition to advantages [58]. For example, the sudden demand for assistant procedures for bleeding or the application of laparoscopic instruments may not be accomplished without a readily available table-side assistant surgeon.

Nevertheless, there is a growing awareness of the fact, that the robotic learning curve in liver surgery is faster than the laparoscopic one and that robotic surgery may allow optimized access to difficult liver segments [51].

Conventional laparoscopic major hepatectomy shows a learning curve of 45 to 75 operations and minor hepatectomy of only 20 to 25 cases, and thus a reduction of operation time [59]. The results of a meta-analysis revealed that robotic major hepatectomy was associated with a longer operative time compared to traditional laparoscopic major hepatectomy. This difference in the longer operative time of RLR was said to be due to docking and undocking of the robot and exchanging of instruments, and partly due to the learning curve. The learning curve for RLR was 30 patients vs. 60 patients in LLR [60]. The study by Zhao et al. suggests that the learning curve of robotic resection in postero-superior segments of the liver might be shorter, based on their experience and previous publications. After 30 robotic liver operations, the operative time is reduced but then plateaued. Conversion rate to open decreased by 4.5% after 70 operations [61]. Zwart et al. found a reduction in operative time and conversion to open after 55 operations and a second reduction point after 145 cases for conversion rate [62].

Tsung et al. performed 57 consecutive robotic hepatectomies and showed significant improvement in surgical parameters in the late phase of the learning curve [63].

A single-center study (140 cases) looked at the conversation to open surgery during the learning phase and found improvement in the conversion rate after 30 cases compared to 60 cases for laparoscopic hepatectomy. This confirms the Morioka consensus statement in which learning MIS is easier with the robotic system [64].

Efanov et al. compared the learning curves of robotic and laparoscopic hepatectomies in a series including 131 MIS hepatectomies performed by 2 MIS inexperienced surgeons. They stated that the learning curve of posterosuperior segment resection in LLR takes only 16 procedures and 29 for LLR. The overall difficulty index significantly increased in the RLR group, in contrast to the laparoscopic group [45].

Advanced robotic surgical procedures can be done during the intermediate phase of the learning curve.

The level of difficulty increased in robotic resections during the learning curve (after 16 cases) and the closeness to major vascular structures and the size of the tumor were higher in the robotic group. In contrast, the level of difficulty did not change in the LLR group.

Other studies concluded that more challenging procedures were performed with RLR over time, while there was no increase in complexity over time in LLR [45].

In another study, there was a significant reduction in operative time in the last 18 robotic hemihepatectomies compared to the first 18 robotic cases (467.9 ± 108.0 min vs. 331.9 ± 129.1 min, *p* = 0.002). The authors argued that the curve for RLR might be flatter if the surgeon already has high experience in LLR. Furthermore, they stated that hilar dissection, hemostasis, and vascular control are easier in RLR, while the instruments are not adequate for parenchymal dissection [65].

O’Connor et al. showed superior outcomes with robotic-assisted minor hepatectomy compared with laparoscopic surgery after 25 cases [66]. Magistri et al. found that operation time reduced significantly after 30 cases of robotic hepatectomy [67].

ICG use and better visualization offered by the Da Vinci robotic system have been said to explain the faster learning curve [52,55] as it can be used with technical advantages in robotics in a particularly user-friendly way.

## 8. Technical and Surgical Aspects

The initial phase of Robotic Liver Surgery consists of patient positioning. The patient is placed in supine and 20° anti-Trendelenburg positioning, usually with the right arm adjoined, allowing the robot to dock on both the right and left side of the patient. The port placement depends on the localization of the lesion. In general, the camera trocar should be placed 15–20 cm away from the xiphoid. The first assistant trocar is placed as primary access to the abdomen in the right mid-abdomen using the separator technique. After creation of the pneumoperitoneum (10–12 mmHg) and trocar placement, diagnostic laparoscopy can be started. The assistant port is usually placed laterally under the camera port to preserve the motion range of the robotic arms [5].

For both left- and right-sided resections, only minor modifications of trocar placement and instrument positioning are usually sufficient to achieve an optimal setting of the transection plane. This is especially true for caudal approach resections, e.g., posterolateral sectionectomy, where the dissection is performed from caudal to cranial [68]. For right hemihepatectomy, the first three trocars are placed in a horizontal line approx. 15 cm subxiphoid with the center of gravity in the right hemiabdomen. The left, fourth trocar for arm 4 is placed slightly cranially offset to this and the surgical robot is connected. When adding the CUSA (cavitronic ultrasonic surgical aspirator) for resection, the corresponding second assistant trocar is positioned cranially of the DaVinci row.

Motion scaling, rather than tremor filtration, plays the major role in the enhanced accuracy seen in robotic surgical systems. Robotic assistance with motion scaling significantly improved accuracy above laparoscopic instruments alone [69].

Major resections begin with performing ultrasound and mobilization of the liver, followed by control of vascular inflow, as well as outflow control and finally resection of the target parenchyma. Following these, the hepatoduodenal ligament can be encircled to prepare for a pringle maneuver to control inflow.

During right or left hemihepatectomy, initially the corresponding hepatic artery and portal vein are dissected after ligation. After transection, the dissection line is demarcated and resection can be performed along with it. If the liver parenchyma is separated from the larger ducts, the Hemolok clip, titanium clip, or absorbable clip should be clipped. For parenchymal dissection, different devices such as the harmonic scalpel, bipolar energy device, endoscopic CUSA via assistant trocar or ligation beam (Ligasure) can be used [70].

The harmonic scalpel is composed of a vibrating blade and a ‘sealing’ blade. The interaction of these two components allows penetration into tissue with parallel reliable hemostasis. For liver tissue, the vibrating blade is an ideal instrument for sparing vascular structures (analogous to the CUSA). A disadvantage is the missing EndoWrist, so that it can only be used parallel to the dissection line [70].

The CUSA is a dedicated liver parenchymal instrument. The cavitron features make it very delicate on vascular structures. However, there is no robotic adaptation available, and it has to be controlled by an assistant surgeon. In summary, there is no perfect robotic tool for liver resection in the current state of technical development. However, exponential technical development of innovative tools can be expected on this platform.

The EndoWrist^®^ One^TM^ Vessel Sealer (Intuitive Surgical Inc., Sunnyvale, CA, USA) is a fully wristed robotic energy device (60° of articulation in all directions) that is used to seal and cut vessels up to 7 mm in diameter.

Nota et al. could demonstrate that the Vessel Sealer is generally employable for parenchymal bile ducts, portal branches, and veins but a stapler and/or hemolocks are used for inflow/outflow pedicles, major veins, or when larger vascular structures are encountered that are clearly beyond a size that could easily be sealed with a margin within the length of the sealer’s surface at 90 degrees. The authors therefore conclude that the Vessel Sealer is appropriate to seal most vascular structures encountered within the parenchyma of the liver segments [71].

Zwart et al. showed, that the vessel sealer is the second most used device for parenchymal transection in an Pan-European survey including 120 surgeons in 24 countries [62].

## 9. ICG in Robotic Liver Surgery

Indocyanine green (ICG) is a fluorescent dye with few side effects, which was initially used for liver function diagnostics. After intravenous injection, ICG binds rapidly to albumin and lipoproteins, which are taken up by hepatocytes. Non-metabolized excretion takes place via the bile ducts [72,73,74,75].

After systemic application, fluorescent hepatocytes and bile ducts fluoresce within a few minutes under irradiation with near-infrared light. The liver segments can be visualized directly or indirectly. In direct imaging, ICG is introduced into the segment-supplying vessels, usually under ultrasound guidance [75].

In indirect imaging, the segmental branches are ligated and then ICG is applied systemically, which leads to a negative contrast of the liver segments [76]. In both cases, identification of the individual anatomy is possible.

Later, the intraoperative identification of primary liver tumors (hepatocellular carcinoma, cholangiocellular carcinoma) occurred by utilizing ICG fluorescence [77,78]. The impaired bile excretion of these tumors or their surrounding hepatocytes leads to a prolonged accumulation of ICG in these tissues, compared with healthy liver tissue.

Generally, ICG is injected intravenously 3–14 days prior to surgery. Due to the described slowed excretion of the tumor or the peritumoral tissue, fluorescent tumor visualization intraoperatively under irradiation with the corresponding light spectrum is possible.

However, in the case of patients with hepatocellular carcinoma, up to 8% false-positive findings have been described [79]. Ishizawa et al. reported a high sensitivity of 99% identification by ICG imaging in their study of 170 HCC patients [80]. Another limitation is the reduced penetration depth of fluorescent light into the tissue [81]. Deeper tumors are therefore difficult to identify.

ICG accumulation can be visualized using a near-infrared (NIR) camera, which is currently often integrated into laparoscopic or robotic systems. The Firefly™camera (Intuitive, Sunnyvale, CA, USA), integrated in the da Vinci Surgical Systems (Intuitive, Sunnyvale, CA, USA) can easily be used to visualize ICG accumulation.

In open surgery, additional NIR cameras are needed and operation room lights need to be switched off. In laparoscopic surgery, the NIR camera is integrated into some systems. The Firefly™ camera can easily be switched on and off and acquire intraoperative ICG-based images that are directly projected onto the operation field. These may further be merged with intraoperative ultrasound (IOUS) images without changing instruments or monitors. Within this context, regarding hepatobiliary surgery, the robotic platform can potentially overcome the drawbacks of the laparoscopic approach and ICG could facilitate the estimation of the required extent of resection and location of the transection line and thus compensate for the lack of haptics.

## 10. Costs

The financial burden of a surgical procedure has to be taken into consideration due to reimbursement issues. In 2016 Intuitive Surgical Inc. owned 80% of the robotic surgery market [82]. However, their surgical robot patents are expiring from 2020 until 2022. Competitors are therefore expected to enter the market with new innovative products soon.

When analyzing costs for RLR vs. LLR or RLR vs. OLR, it is necessary to distinguish between individual cost factors, total costs, and their statistical significance respectively. Limited data is available to compare subgroups of resection techniques and primary liver patient characteristics.

A meta-analysis by Zhao et al., including 39 papers and 1249 RLR, 1010 LLR, and 740 OLR, comparing operative costs of RLR and LLR, showed significantly higher operative costs of the robotic instrumentation and longer operative time, hospitalizations, and ultimately total costs for RLR. A meta-analysis of total costs for RLR vs. OLR showed no significant difference. Thus, when including maintenance costs of RLR, it is more costly than LLR and OLR [83].

Similarly, Ziogas et al. did a systematic review of 38 papers including 390 RLR, 1674 LLR, and 783 OLR. The economic meta-analyses showed higher total costs for RLR vs. LLR, as well as higher operative and hospitalization costs, but no significant difference in total costs for RLR vs. OLR. Some studies suggest lower hospitalization costs for RLR due to decreased LOS and associated costs [59].

A retrospective study of 71 RLR and 88 OLR at a single institution found that perioperative costs ($6026 vs. $5479; *p* = 0.047) were higher, and operative time was 20% longer in RLR, whereas LOS was 2 days shorter and postoperative costs (room, ICU, laboratory, and miscellaneous inpatient costs) were lower, and thus direct costs compared to OLR ($14,754 vs. $18,998; *p* = 0.001) were lower as well.

Notably, the RLR group was composed of 75.8% minor hepatectomies (1 or 2 Couinaud segments) and 24.2% major (C3 segments) resections. In the OLR group were 48.9% major hepatic resections [84], which led to a limited comparability of the groups.

On the contrary, one study by Sham et al. compared the relative costs of open versus robotic hepatectomy (*n* = 71) and open hepatectomy (*n* = 88) by combining both perioperative and postoperative costs on the overall costs of each operation at a single center over 3 years. Postoperative care costs were decreased by reduced LOS [84].

However, patients chosen for open surgery had a higher rate of major resections, tumors located in the posterior segments of the liver, and physical status classification (ASA class). Second, the additional costs for disposables and maintenance costs were not included [49,85].

In another study, 86 RLR and 55 OLRs were performed at a single institution; 42.7% were major hepatectomies and 57.3% were minor resections in the RLR group performed by one surgeon. In the open group were 43.6% major and 56.4% minor resections performed by five surgeons. The economic parameters comprised of direct variable costs, direct fixed costs, and indirect costs. The authors calculated an average total cost of $37,518 for robotic surgery vs. $41,948 for open surgery [86].

A cost analysis of the total cost of a single-center comparing robotic and open hepatectomy was performed in a study including costs of perioperative (operating room and anesthesia) and inpatient care (laboratory, room, board, complication treatment, examinations, etc.). The purchase price of the da Vinci^®^ Surgical System and its associated annual maintenance fee were not included. In the analysis, robotic surgery showed longer operation time and thus higher perioperative costs. On the contrary, robotic liver resection (RLR) resulted in lower LOS and associated costs, thus inpatient care costs was lower compared to open resection. Taken together, robotic surgery appeared to be more costly than open hepatectomy. Interestingly, the authors state that in Taiwan, costs for inpatient care are lower, because of the national reimbursement system, thus overshadowing the cost-benefit of shorter LOS, negatively affecting the total costs of RLR compared to nations spending more money on inpatient care [87]. In Germany, the healthcare system predefines the postoperative stay for reimbursement, thus influencing total costs [88].

## 11. Discussion

Robotic-assisted liver surgery is a highly promising new technique for overcoming the limits of minimally invasive liver resection. It is safe and feasible in the hands of an experienced hepatobiliary surgeon. The available retrospective studies indicate R0/R1 resection rate, overall and recurrence-free survival rates to be similar in laparoscopic and robotic resections.

It is noteworthy that the current evidence is based solely on retrospective and uncontrolled case series, and thus it seems rather ambitious to speak of reliable evidence. However, randomized studies need to validate these outcomes by comparing cases with similar complexity. This is also important to lower the biases in the literature.

In addition, randomized controlled studies are urgently needed to validate and conclude indications and contraindications for certain hepatic resection maneuvers. Nowadays there are no other definite contraindications for RLR than for OLR, and at the end of the day, it is an adjusted decision of the surgeons if they are confident to perform a robotic resection in even complex situations. An experienced team can achieve the same results as with LLR and OLR while the patient is not put at a higher risk compared to the alternative.

Since HCC often occurs in cirrhosis, MIS is preferred for liver resection as there is a higher risk of complications leading to longer LOS and mortality. CCC is often associated with large tumors, a multifocal distribution pattern, and lymph node metastases, making MIS challenging and thus is only recommended in the non-cirrhotic liver and without lymphadenectomy. CCC is very rare and evidence for this subgroup is hard to find in the literature. However, from small case series, MIS is suggested to be effective in terms of blood loss and technical feasibility. There is a strong need to prove the feasibility and effectiveness of MIS in CCC. Randomized trials of laparoscopic versus open liver surgery for HCC are missing, meta-analyses comparing MIS to open provide the best evidence so far, showing better or equal effectiveness for parameters for MIS resections with the exclusion of operation time.

There are even fewer studies comparing robotic-assisted laparoscopy and traditional laparoscopy. As mentioned earlier, crystallizing the benefits of robotics by the usual parameters is questionable here. However, meta-analyses from existing field reports confirm similar effectiveness to laparoscopy and superior effectiveness especially in highly complex resections around the hilus, making robotic surgery a great improvement for this subgroup. The reason for better performance in complex surgeries is that the robotic system offers some technical advantages over laparoscopic instruments. Surgical systems have been improved over time and shortly new companies are expected to bring innovative surgical products onto the market, further enhancing the technological advantages of robotic surgery.

Regarding the learning curve, again most studies are retrospective and the results of those studies depend on the difficulty of the hepatectomy, the version of the robotic system used, and the individual experience of the surgeon, making it difficult to compare. Furthermore, most studies are retrospective in nature.

The learning curve of RLR, which is found to be faster than in laparoscopic resection, can be divided into three main phases and a stepwise approach is recommended. Over time there is an observed improvement of surgical outcomes seen in a reduction of operating time, intraoperative complications, and LOS, and the learning curve is plateaued with no further improvement. Thus, after passing the learning curve, RLR can be considered safe and feasible especially for anatomically difficult liver segments. Overall, a center achieves the best results when the surgeon before starting the learning curve program shows extensive experience in liver anatomy, open resection, and laparoscopic resection.

Concerning economics, the literature does not provide comparable parameters in the studies. Some studies did for example not include the purchase and maintenance costs of the robotic system while others did. When considering these high purchase costs alone one can conclude that generally, a higher volume of robotic-assisted surgeries leads to better cost-effectiveness. This can explain why small volume centers could be more reluctant to obtain the surgical system from an economical point of view. Passing the learning curve helps to achieve a better cost-effectiveness, as operation time is decreasing and surgical outcomes are improving. Further, differences between the results of costs in the studies may reflect the difference in reimbursement systems in different countries.

## 12. Conclusions and Future Perspectives

Our literature review indicates that robotic resections of primary liver tumors are safe, feasible, and have a low conversion rate. Nevertheless, it must be noted that the existing evidence is based primarily on retrospective uncontrolled case studies and thus there is an imperative need for randomized studies to investigate the topic scientifically in a suitable approach and to enable conclusions.

Comparing robot-assisted resections with open liver resections, a longer operation time for the robotic approach is described in almost all publications. However, the intraoperative blood loss is lower, the perioperative complication rate is reduced and the postoperative hospital stay is shorter than with conventional open procedures. Over the past 20 years, MIS has had an increasing role in liver surgery. When comparing the laparoscopic and robotic approaches, those differences cannot be displayed so evidently. Nevertheless, robotic liver surgery will allow surgeons to conduct more complex resections and to complete more procedures purely minimally invasively. It is now thoroughly established that robotic liver surgery is safe and feasible compared to open and laparoscopic surgery, with at least non-inferior oncological outcomes.

This approach especially facilitates dissection of the hepatic hilum and hepatocaval plane, mobilization of the liver attachments, biliary anastomosis, and suturing for bleeding management during the parenchymal transection and reconstruction.

Furthermore, the robotic platform with promising new technical innovations, allows for easier integration, such as near-infrared fluorescence for vascular and biliary identification. Augmented reality, image-guided surgery, and 3D ultrasound instruments with integrated probes for section margin assessment are all implementations that soon will make resections safer and more efficient, and contribute to the robotic approach being used as a standard surgical approach to liver malignancies.

In addition, there is robust evidence for a less steep learning curve for robotic liver surgery compared to laparoscopic surgery, which will have a great impact on future results.

Although considering that the horizon of experience in robotic-assisted liver surgery is equivalent to that of laparoscopic liver surgery approximately 25 years ago and that the first prospective randomized studies in laparoscopic liver surgery were published only recently, no valid long-term results for robotic-assisted liver surgery can be expected in the near future.

Nevertheless, the authors believe that equal R0 rates and identical lymph node rates in combination with reduced blood loss and enhanced recovery will translate into superior oncological results in future trials evaluating robotic liver surgery.

Lastly, there are no large, prospective studies regarding robotic hepatectomies especially for primary liver neoplasms published to date. Prospective and multicenter trials are needed to validate the current promising results.

## Figures and Tables

**Table 1 cancers-14-00265-t001:** Surgical outcome of CCC patients after liver resection.

References	Patients (n)	Surgical Resection Method	Mean Intraoperative Blood Loss in mL	Length of Hospital Stay (LOS); Mean Postoperative Stay	Conversion Rate in %	Complication Rate in %	Mean Operative Time in Min
CCC							
Ratti et al. 2016 [8]	20	Laparoscopic resection	200	2–10	5	15	210
	60	Open resection	350	3–21	-	13	180
Xu et al. 2016 [9]	10	Robotic-assisted resection	1360	16	n.a.	90	703
	32	Open resection	1014	14	-	50	475
Sucandy et al. 2021 [10]	15	Robotic-assisted resection	150	4	0	0	453

**Table 2 cancers-14-00265-t002:** Surgical outcome of HCC patients after liver resection.

References	Patients (n)	Surgical Resection Method	Mean Intraoperative Blood Loss in mL	Length of Hospital Stay (LOS); Mean Postoperative Stay	Conversion Rate in %	Complication Rate in %	Mean Operative Time in Min
HCC							
Lai et al., 2013 [11]	41	Robotic-assisted resection	413	6.2	47%	7.1%	229
Wang et al. 2017 [12]	63	Robotic- assisted resection	206	6.21	0%	11%	296
	177	Open resection	267	8.18	-	15.3%	182
Chen et al. 2017 [13]	81	Robotic-assisted resection	282	7.5	1.6%	4.9%	343
	81	Open resection	263	10.1	-	4.9%	220
Lim et al. 2020 [14]	49	Laparoscopic resection	n.a.	7	14%	27%	269
	44	Robotic-assisted resection	n.a.	9	5%	16%	252
Kato et al. 2020 [15]	57	Robotic- assisted resection	194	15	2%	11%	612

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
