# Peer review of "Robotic-Assisted Surgery for Primary Hepatobiliary Tumors—Possibilities and Limitations"

_cancers, 2022, doi:10.3390/cancers14020265_

Round 1

Reviewer 1 Report

I read with interest the narrative review by Spiegelberg, J et al. entitled: "Robotic-assisted surgery for primary hepatobiliary tumors - possibilities and limitations".

The article is interesting and well written even if it has the limit of not being a systematic review. However, the article is too long and I think it can be shortened (several concepts are repeated).

I have also highlighted some minor revisions

Introduction

Line 61. The most recent Da Vinci model is the Xi and not the X

Methods

Line 83. The word cholangiocarcinoma was not included in the research.

Line 87. Until or since?

Cholangiocellular carcinoma

Line 138: “LAD” acronym needs to be written extensively

Line 144: trisectorectomy -> trisectionectomy

Technical considerations

Line 238: Indocyanine green also exists for laparoscopy

Indication / Contraindication

Line 309: OCC?

Line 324: RS?

Learning Curve

Line 355: substitute "hospital bed length" with "hospital stay" or "length of stay"

Line 384: the term "artificial intelligence" is wrongly used in this context, and the article herein referenced (Favre, Angeline et al. "Pedagogic Approach in the Surgical Learning: The First Period of" Assistant Surgeon "May Improve the Learning Curve for Laparoscopic Robotic-Assisted Hysterectomy. ”Frontiers in surgery vol. 3 58. 2 Nov. 2016, doi: 10.3389 / fsurg.2016.00058) has a totally different topic: by no means the table surgeon minimizes the role of the robotic system.

Line 439: Indocyanine green also exists for laparoscopy

Technical and surgical aspects

Line 483-501: Is it really necessary to describe in detail a procedure in a review article? Maybe it's better to just expose all the techniques and instruments singularly.

Line 505: CVD acronym is not explained

ICG in robotic liver surgery

Indocyanine green also exists for laparoscopy and for the open technique. The differences in the use / effectiveness of greenery in the three approach techniques should possibly be indicated

Line 533: ICG, not IGC

Discussion

To date, no significant advantages of robotic surgery over laparoscopic have been highlighted for the patient. For the surgeon, on the other hand, there can be several advantages (a lower learning curve, a better operative well-being). Future studies should be aimed precisely in this area

It would be useful to describe the advantages-disadvantages of the use of intraoperative ultrasound in robotic surgery

I would put the limits on the use of the CUSA in hepatic robotic surgery

Figure 1

In my opinion, Figure 1 is not adequate. I think that intraoperative picture is better

Reviewer 2 Report

This is an interesting narrative review that aims to describe to potential advantages and pitfalls of robotic surgery for primary liver tumors (HCC and cholangiocarcinoma). While overall this a timely subject worth nuanced discussion, the manuscript is now overly comprehensive and would benefit from (much) more focus on its primary subject.

Specific comments

  • Abstract: I respectfully disagree that there is “high controversy about the surgical feasibility” of robotic surgery. In fact, there are overwhelming amounts of clinical data from studies in urology, gynecology, and general/GE surgery showing the feasibility of the robotic approach. Robotics is nowadays widely employed in clinical practice in many countries.
  • Cholangiocarcinoma: this section would benefit from a concise description of the technical advantages of the robotic approach (e.g., easier hepatic pedicle dissection, hilar dissection etc) allowing wider application of MIS in relation to patient benefit of the MIS approach (enhanced recovery – this, at least, was shown for standard hepatectomies in the OSLO-COMET and ORANGE trials). For extended / complex resections, e.g., Klatskin tumors, potential patient benefit may be less clear. Larger series of robotic Klatskin resections should be included (eg., Sucandy et al. J Surg Oncol ’21).
  • HCC: a discussion of the literature on robotic hepatectomy for primary liver cancer is lacking? E.g., Khan et al. Ann Surg Oncol ’18.
  • Robotic resection:
    • although this is a narrative review, a table summarizing published series on (or including/specifying) robotic resection of iCCA/pCCA/HCC would be very clarifying. Gallbladder cancer should be included as well.
    • more than once it is suggested that randomized studies are needed. I would approach this point with care: randomization between lap and robot is not feasible (surgeons would have to go through learning curve of both techniques…) and, in the opinion of many, not mandated as in fact both are variants of MIS with differential technical capabilities. Randomization between open and lap. hepatectomy has been done, clearly showing the advantage of MIS per se.
  • Technical aspects:
    • add scaling of movement to potential advantages of robotic surgery.
    • Fluorescence is also available in laparoscopic surgery ; the fact that the surgeon can control optical switching in the console (using Firefly mode) is specific to the robot, as is probe control and in-console projection of robotic intra-operative US.
    • Please skip the description and schematic of robotic right hepatectomy here, it is not specific for the subject of the review and makes the ms unnecessarily long.
    • Methods of parenchymal transection may be discussed here, and should include the vessel sealer (which is the 2nd most used device for parenchymal transection (Zwart et al. HPB ’21).
    • Consider to shorten the paragraph on learning curve and move it to the discussion, as this is not specific to primary liver tumors.
  • Discussion: would benefit from shortening and focus on the primary subject (robotics in resection of primary liver cancer).

Round 2

Reviewer 2 Report

the manuscript has been adapted and improved, i recommend to accept it!